# The Efficacy of Targeted Monoclonal IgA Antibodies Against Pancreatic Ductal Adenocarcinoma

**DOI:** 10.3390/cells14090632

**Published:** 2025-04-24

**Authors:** Léon Raymakers, Elsemieke M. Passchier, Meggy E. L. Verdonschot, Mitchell Evers, Chilam Chan, Karel C. Kuijpers, G. Mihaela Raicu, I. Quintus Molenaar, Hjalmar C. van Santvoort, Karin Strijbis, Martijn P. W. Intven, Lois A. Daamen, Jeanette H. W. Leusen, Patricia A. Olofsen

**Affiliations:** 1Center for Translational Immunology, University Medical Center Utrecht, Heidelberglaan 100, 3584 CX Utrecht, The Netherlands; l.raijmakers-3@umcutrecht.nl (L.R.); e.m.passchier-3@umcutrecht.nl (E.M.P.); m.verdonschot@amsterdamumc.nl (M.E.L.V.); j.g.m.evers-4@umcutrecht.nl (M.E.); p.a.olofsen@umcutrecht.nl (P.A.O.); 2Division of Imaging & Oncology, University Medical Center Utrecht Cancer Center, Heidelberglaan 100, 3584 CX Utrecht, The Netherlands; m.intven@umcutrecht.nl (M.P.W.I.); l.a.daamen-3@umcutrecht.nl (L.A.D.); 3Department of Pathology, Regional Academic Cancer Center Utrecht, UMC Utrecht Cancer Center & St. Antonius Hospital Nieuwegein, St. Antonius Hospital Nieuwegein, Koekoekslaan 1, 3435 CM Nieuwegein, The Netherlandsg.raicu@antoniusziekenhuis.nl (G.M.R.); 4Department of Surgery, Regional Academic Cancer Center Utrecht, UMC Utrecht Cancer Center & St. Antonius Hospital Nieuwegein, Utrecht University, Heidelberglaan 100, 3584 CX Utrecht, The Netherlands; i.q.molenaar@umcutrecht.nl (I.Q.M.); h.van.santvoort@antoniusziekenhuis.nl (H.C.v.S.); 5Department of Biomolecular Health Sciences, Division Infectious Diseases and Immunology, Faculty of Veterinary Medicine, Utrecht University, Yalelaan 1, 3584 CL Utrecht, The Netherlands; k.strijbis@uu.nl

**Keywords:** IgA, monoclonal antibodies, mAbs, neutrophils, polymorphonuclear cells, PMNs, pancreatic cancer, pancreatic ductal adenocarcinoma, PDAC

## Abstract

The efficacy of immunotherapy in pancreatic ductal adenocarcinoma (PDAC) remains limited. The tumor microenvironment (TME), characterized by the accumulation of suppressive myeloid cells including neutrophils, attributes to immunotherapy resistance in PDAC. IgA monoclonal antibodies (mAbs) can activate neutrophils to kill tumor cells; this can be further enhanced by blocking the myeloid immune checkpoint CD47. In this study, we investigated the potential of this therapeutic strategy for PDAC. We determined the expression of tumor-associated antigens (TAAs) on PDAC cell lines and fresh patient samples, and the results showed that the TAAs epithelial cell adhesion molecule (EpCAM), trophoblast cell surface antigen 2 (TROP2) and mucin-1 (MUC1), as well as CD47 were consistently expressed on PDAC. In line with this, we showed that IgA mAbs against EpCAM can activate neutrophils to lyse various PDAC cell lines and tumor cells, which can be augmented by addition of CD47 blockade. In addition, we observed that neutrophils were present in patient tumors and expressed the receptor for IgA. In conclusion, our results indicate that a combination of IgA mAb with CD47 blockade is a promising preclinical treatment strategy for PDAC, which merits further investigation.

## 1. Introduction

Pancreatic ductal adenocarcinoma (PDAC) remains one of the most lethal cancers, with a 5-year survival rate of only 13% for all disease stages combined [1]. Given that 40% of patients experience locally advanced disease and 40% of patients have distant metastasis at primary diagnosis, only 15–20% of patients are eligible for tumor resection [2]. Despite advancements in chemotherapeutic regimens and surgical approaches, major therapeutic breakthroughs have not been forthcoming [3]. In the context of its rising incidence, PDAC is expected to become the second most common cause of cancer-related death by 2030 [4,5]. Therefore, there is a need for new innovative treatment strategies for PDAC.

Monoclonal antibodies (mAbs) can activate the immune system to fight cancer cells, and this type of therapy includes immune checkpoint inhibitors and antibodies targeting tumor antigens (targeted mAb therapy) [6]. Immune checkpoint inhibitors block certain protein interactions which inhibit the (anti-tumor) immune response, thereby reactivating previously suppressed immune cells [6]. Immune checkpoint inhibitors have substantially improved outcomes in several cancer types [7,8,9,10]; however, this has not been achieved for PDAC [11,12,13,14]. Targeted mAbs binding to tumor antigens can have both direct and indirect effects, e.g., blocking growth factor receptor signaling or inducing cancer cell killing via complement-dependent cellular cytotoxicity, antibody-dependent cellular cytotoxicity (ADCC), and antibody-dependent phagocytosis [6]. Targeted mAb therapy with cetuximab (directed against the epidermal growth factor receptor/EGFR) was tested for PDAC in phase III clinical trials but failed to improve overall survival while adding toxicity [15,16].

The inefficacy of immunotherapies in PDAC could be attributed to its highly immunosuppressive tumor microenvironment (TME), which is characterized by a dense desmoplastic stroma and accumulation of suppressive myeloid cells, including neutrophils [3,17,18,19,20,21]. Compared with other immune cell subsets, neutrophil infiltration is associated with the poorer outcomes in cancer [22]. In line with this, neutrophil infiltration is associated with poor prognosis in PDAC [19,23,24]. Neutrophils often have a pro-tumor phenotype in the TME of PDAC and are associated with tumor growth, angiogenesis, fibrosis, and reduced efficacy of therapy [19,25,26]. Furthermore, neutrophils can exert immunosuppressive functions by inhibiting the T cell and NK cell anti-tumor immune response [19,25]. Consequently, research efforts have focused on neutrophils, aiming to interfere with their immunosuppressive effector functions, reduce neutrophil infiltration, or deplete neutrophils altogether [19,25,26,27,28,29,30]. To date, these therapies lack feasibility for clinical practice.

Alternatively, neutrophils can be activated to exhibit anti-tumor effects via the use of mAbs directed against tumor-associated antigens (TAAs) [31,32,33,34]. Antibodies of the IgA isotype have been shown to be particularly effective in activating neutrophils compared with their respective IgG counterparts [31]. Neutrophils comprise 50–70% of all leukocytes in the peripheral blood and are abundantly present in PDAC tumors [19,20]. Due to the immunosuppressive roles of neutrophils, activating neutrophils to execute an anti-tumor effect could also boost T cell and NK cell anti-tumor immunity [19,25,35]. Therefore, targeted IgA mAbs pose a promising therapeutic strategy for PDAC. However, tumors can mitigate IgA-mediated killing by expression of myeloid immune checkpoints such as CD47, which binds to signal regulatory protein alpha (SIRPα) on myeloid cells and thereby counteracts the activation signal provided by mAb binding [36,37]. Especially in cancer with high CD47 expression, additional blockade of the CD47/SIRPα axis has been shown to enhance IgA mAb efficacy [36,38].

In this study, we characterized the expression of TAAs on PDAC cell lines and patient tumor samples and tested the efficacy of IgA mAbs directed against these TAAs, to investigate whether IgA mAbs in combination with CD47 blockade could pose a promising therapeutic strategy for PDAC.

## 2. Materials and Methods

An overview of all materials, reagents, and antibodies is presented in Appendix A.

### 2.1. Cell Lines

All tumor cell lines were purchased from the American Type Culture Collection (ATCC) and cultured at 37 °C in a humidified incubator with 5% CO_2_. AsPC-1 (CRL-1682. RRID:CVCL_0152), BxPC-3 (CRL-1687, RRID:CVCL_0186), and Panc 10.05 (CRL-2547, RRID:CVCL_1639) cell lines were cultured in RPMI 1640 with Hepes and GlutaMAX (Thermo Fisher, Gibco, Grand Island, NY, USA, catalog number (cat#) 72400-021) supplemented with 10% heat-inactivated fetal calf serum (FCS, Sigma-Aldrich, St. Louis, MO, USA, cat# F7524) and 100 U/mL penicillin–streptomycin (Pen/Strep, Fisher Scientific, Gibco, Waltham, MA, USA, cat# 11548876). CFPAC-1 (CRL-1918, RRID:CVCL_1119) cells were cultured in IMDM (Thermo Fisher, Gibco, cat# 12539089) and Capan-2 (HTB-80, RRID:CVCL_0026) in McCoy’s 5A (Thermo Fisher, Gibco, cat# 15410604), both supplemented with 10% FCS and 100 U/mL Pen/Strep. Cell lines were passaged with trypsin-EDTA (10× diluted 1:10 in phosphate buffered saline/PBS, Thermo Fisher, Gibco, cat# 15400-054) according to ATCC guidelines and were not cultured past 30 passages. The tumor cells lines were tested frequently for mycoplasma, using a Mycoalert mycoplasma detection kit (Lonza, Basel, BS, Switzerland, cat# LT07-318).

### 2.2. Patient Samples and Processing

Residual patient samples were acquired from treatment-naïve patients who underwent surgery for primary PDAC within the Regional Academic Cancer Center Utrecht (RAKU) and did not object to the use of anonymous body material under the Medical Treatment Contracts Act or WGBO. A fresh tumor sample from the resected specimen was obtained directly after surgery via macroscopic resection by the pathologist. The anonymized tumor sample was immediately processed and analyzed within hours after the surgery. The patient tumor samples were transferred to the lab in tubes with RPMI 1640 or MACS Tissue Storage Solution (Miltenyi Biotec, Bergisch Gladbach, NRW, Germany, cat# 130-100-008) and kept on ice during transport. The fresh tumor samples were minced into fragments < 4 mm, using a scalpel. This was followed by enzymatic dissociation with the Human Tumor Dissociation Kit (Miltenyi Biotec, cat# 130-095-929) and gentleMACS Octo Dissociator with heaters (Miltenyi Biotec, cat# 130-096-427) according to the manufacturer’s instructions. The single-cell suspension was analyzed with flow cytometry both before and after red blood cell (RBC) lysis (Figure 1). RBCs were lysed with 10× RBC lysis buffer (Biolegend, San Diego, CA, USA, cat# 420302) diluted tenfold in water. Cells were incubated for 10 min at 4 °C and then washed with PBS. After RBC lysis, the tumor cells were enriched for ADCC experiments via negative isolation through magnetic cell separation (Miltenyi Biotec, cat# 130-092-857 & cat# 130-042-303) according to the kit manufacturer’s instructions. Negative selection was performed with anti-phycoerythrin (PE) microbeads (Miltenyi Biotec, cat# 130-048-801) and LD columns (Miltenyi Biotec, cat# 130-042-901) after labeling with antibodies against CD45-PE (clone HI30, BD Pharmigen, Franklin Lakes, NJ, USA, cat# 555483, dilution 1:200), CD31-PE (clone WM59, Biolegend, cat# 303106, dilution 1:25), PDGFRα-PE-cyanine7 (PE-Cy7)(CD140a, clone 16A1, Biolegend, cat# 323507, dilution 1:50), and PDGFRβ-PE (CD140b, clone 18A2, Biolegend, cat# 323605, dilution 1:50). The purity of the sample was assessed with flow cytometry.

The resected specimens were transferred to pathology department. A sample of untreated tumor was obtained and transferred to the lab in tissue transfer solution. Definitive diagnosis of the tumor took roughly a week and was unclear at the time of processing. The samples were further processed via mechanical and enzymatic dissociation. Flow cytometry was performed to test for the presence of immune cells and tumor-associated antigens (TAAs). Tumor cells were isolated via magnetic depletion from the remaining single-cell suspension, and antibody-dependent cellular cytotoxicity (ADCC) assay was performed.

### 2.3. Antibody Production

Engineered IgA3.0 and IgG1 antibodies, herein referred to as IgA and IgG antibodies, against EGFR (cetuximab), human epidermal growth factor receptor 2 (HER2, trastuzumab), GD2 (ch14,18), annexin (ch2448), mesothelin (3C2), CD70 (41D12), EpCAM (heING1), folate receptor 1 (FOLR1, STRO-002), and TROP2 (sacituzumab), were produced in-house as previously described [32,39]; naked formats were used. To block the CD47/SIRPα myeloid immune checkpoint (herein referred to as CD47 blocking), high-affinity IgG1 LALAPG SIRPα fusion protein (consensus variant 1) was produced and purified in-house [38,40,41]. The IgG1 antibody (humanized 5E5) against Tn/STn MUC1 epitopes (CIM301-1) was kindly gifted by Wilfred T. V. Germeraad (Maastricht University Medical Center, CiMaas). The mouse IgG1 antibodies against MUC1 (214D4 & 139H2), MUC4 (clone 8G7, Santa Cruz Biotechnology, Dallas, TX, USA, cat# sc-53945), and MUC16, were provided by Karin Strijbis (Utrecht University).

### 2.4. TAA Detection

Antibody binding was tested with human(ized) IgA antibodies or mouse IgG antibodies on a panel of potential immunotherapy targets, based on the literature and the available IgA antibodies in house [4,7,42,43,44,45,46]. Flow cytometry analysis was used to determine TAA expression. For cell lines, a total of 100,000 cells were stained with antibody in FACS buffer (PBS (pH 7.4), 0.5% BSA, Roche diagnostics, Basel, BS, Switzerland, cat# 10735094001; 0.1% NaN3 sodium azide, Thistle scientific, Rugby, Wa, UK, cat# SBL-40-2010-01) or PBS for 45 min on ice. For patient tumor samples, preferably 100,000 cells but a minimum of 10,000 cells were stained and high-expressing targets were preferentially measured in case of low cell numbers. For indirect staining, cells were stained with 10 µg/mL unconjugated human IgA or IgG or mouse IgG. After washing twice with PBS, binding was detected with 10 µg/mL PE-conjugated goat anti-human IgA (F(ab’)2 (SouthernBiotech, Birmingham, AL, USA, cat# 2052-09), PE-conjugated goat anti-human IgG (F(ab’)2 (Southern Biotech, cat# 2042-09) or APC-conjugated anti-mouse IgG (H + L) F(ab’)2 (Southern Biotech, cat# 1032-11) antibody by incubating for 45 min on ice in the dark. Direct staining was performed with fluorescent labeled antibodies at proper dilutions for 45 min on ice in the dark. PE-conjugated antibodies against EGFR (clone AY13, Biolegend, cat# 352903, dilution 1:100), EpCAM (clone 9C4, Biolegend, cat# 324206, dilution 1:25), TROP2 (clone NY18, Biolegend, cat# 363803, dilution 1:25), MUC1 (clone 16A, Biolegend, cat# 355603, dilution 1:25), HER2 (clone 24D2, Biolegend, cat# 324405, dilution 1:25), and FOLR1 (clone LK26, Biolegend, cat# 908303, dilution 1:25) were used for direct TAA staining. CD47 was detected with a PE-conjugated antibody (clone CC2C6, Biolegend, cat# 323108, dilution 1:50). Dead cells were excluded using 7-Amino-Actinomycin D (7-AAD; BD, cat# 559925, dilution 1:20) or To-Pro-3 (Thermo Fisher, Invitrogen, Waltham, MA, USA, cat# T3605, dilution 1:50,000). A FACS Canto II flow cytometer (BD Biosciences, Franklin Lakes, NJ, USA) was used to perform the measurements.

### 2.5. Quantification of TAA Expression

QIFIKIT (Agilent, Dako, Santa Clara, CA, USA, cat# K007811-8) analysis was performed to determine the antigen density (numbers of molecules per cell). Staining was performed according to the manufacturer’s instructions with 10 µg/mL unconjugated monoclonal mouse IgG antibody directed against EpCAM (9C4, Biolegend, cat# 324201), EGFR (AY13, Biolegend, cat# 352901), TROP2 (TACSTD2, MR54, Invitrogen, cat# 14-6024-82), or CD47 (CC2C6, Biolegend, cat# 323102). We did not perform this analysis for MUC1, since MUC1 antibodies have multiple binding sites per molecule and QIFIKIT would therefore not accurately reflect the number of molecules per cell.

### 2.6. Analysis of Immune Cell Subsets

Immune cell subsets in dissociated patient tumors were analyzed via flow cytometry after RBC lysis. A minimum of 200,000 cells were stained with fluorescent antibodies as described above. Immune cell subsets were characterized using CD45-brilliant violet 510 (BV510) (clone HI30, Biolegend, cat# 304036, dilution 1:200), CD66b-Alexa Fluor 647 (AF647) (clone G10F5, Biolegend, cat# 305110, dilution 1:400), CD89-PE (clone A59, BD, cat# 555686, dilution 1:50) CD14-fluorescein isothiocyanate (FITC) (clone TÜK4, Miltenyi, cat# 130-080-701, dilution 1:50), CD3-pacific blue (PB) (clone UCHT1, BD Pharmingen, cat# 558117, dilution 1:20), CD20-allophycocyanin hilite 7 (APC-H7) (clone 2H7, BD, cat# 560734, dilution 1:100), and CD56-PE-Cy7 (NCAM 16.2, BD, cat# 335826, dilution 1:200) antibodies. Further characterization of myeloid cells was carried out using CD66b-AF647, CD68-FITC (Y1/82a, Biolegend, cat 333806, dilution 1:50), CD14-APC-H7 (MφP9, BD, cat# 560180, dilution 1:40), HLA-DR-PB (L243, Biolegend, cat# 307624, dilution 1:100), CD89 peridinin chlorophyll protein cyanine5.5 (PerCP-Cy5.5) (A59, BD, cat# 555686, dilution 1:80) and LOX-1-PE (15C4, Biolegend, cat# 358604, dilution 1:50) antibodies. Lymphocytes were further characterized using CD3-PB, CD4-APC (RPA-T4, Biolegend, cat# 300514, dilution 1:100), CD8-PE (RPA-T8, Biolegend, cat# 301008), CD25-PerCP-Cy5.5 (BC96, Sony Biotechnology, San Jose, CA, USA, cat# 2113130, dilution 1:100), FoxP3-alexa fluor 488 (AF488) (PCH101, eBioscience, San Diego, CA, USA, cat# 53-4776-73, dilution 1:50), CD20-APC-H7 and CD56-PE-Cy7 antibodies.

### 2.7. ADCC Assay

Chromium-51 (51Cr) release ADCC assays were performed as previously described [47]. In brief, the targets cells, either cancer cell lines or isolated patient tumor cells, were labeled with 100 µCi chromium-51 (PerkinElmer) per 1,000,000 cells in a humidified incubator at 37 °C and 5% CO_2_ for a minimum of two hours. Following labeling, cells were washed three times with complete RPMI medium (RPMI supplemented with 10% FCS + 100 U/mL Pen/Strep). For additional blockade of the CD47/SIRPα axis, target cells were incubated with 10 µg/mL IgG1 LALAPG SIRPα fusion protein for 30 min at room temperature. Peripheral blood was obtained from healthy donors within the UMC Utrecht. Human polymorphonuclear leukocytes (PMNs) were isolated from peripheral blood by performing a Ficoll (GE Healthcare, Chicago, IL, USA, cat# GE17-1440-03) density gradient centrifugation. The plasma and peripheral blood mononuclear cells fractions were removed, leaving the PMN and RBC in pellet. RBC were lysed using RBC lysis buffer (Biolegend, San Diego, CA, USA, cat# 420302) leaving the PMN fraction. Cells were resuspended in complete medium. Whole leukocytes (WL) were isolated from peripheral blood by performing RBC lysis and resuspended in complete medium corresponding to the original volume of blood. Antibody dilutions and controls were made with complete medium and added to a round bottom 96 wells plate (Fisher Scientific, Corning, cat# 3799). 5000 target cells were added per well, followed by the addition of effector cells. The effector-to-target (E:T) ratio used for PMNs was 40:1 and WL were diluted 4 times when added to the plate. Cells were incubated in a humidified incubator at 37 °C and 5% CO_2_ for 4 h. After incubation, plates were centrifuged and supernatant was transferred to a LumaPlate (Revvity, Waltham, MA, USA, cat# 6006633) to be measured on a beta-gamma counter for radioactive scintillation in counts per minute (CPM) (PerkinElmer). Background/basal CPM were determined by measuring targets cells without antibodies and effector cells. Maximal CPM were determined by treating target cells with 5% Triton-X-100 (Sigma-Aldrich, cat# X100). Specific lysis was calculated using the formula: (CPM − background CPM)/(maximal CPM − background CPM) × 100%.

### 2.8. Data Processing and Statistical Analyses

Flow cytometry analysis was performed in FlowJo v10 (TreeStar, Woodburn, OR, USA). Statistical analyses were performed using GraphPad Prism 10.1.2 (GraphPad Software Incorporated, San Diego, CA, USA) and represented as mean ± standard error of the mean (SEM) with a *p*-value of <0.05 considered statistically significant. The statistical tests used and *p*-values can be found in the figures. Number of experiment repeats and number of pooled experiments are reported in the figure legends.

## 3. Results

### 3.1. IgA Antibodies Against Tumor-Associated Antigens (TAAs) Can Activate Neutrophils to Induce ADCC of PDAC Cell Lines

The expression of the TAAs EGFR, HER2, GD2, Annexin, Mesothelin, CD70, EpCAM, FOLR1, TROP2, MUC1, MUC4 and MUC16 was assessed on five PDAC cell lines (AsPC-1, BxPC-3, Capan-2, CFPAC-1 and Panc 10.05) (Figure 2A–C and Appendix A). GD2, Annexin, Mesothelin, CD70 and MUC16 were not expressed by all cell lines while expression of HER2, FOLR1 and MUC4 was low and therefore not further evaluated (Appendix A). EpCAM was the most consistent and highest expressed TAA followed by MUC1, TROP2 and EGFR (Figure 2A,B). In addition, expression of CD47 was found on all cell lines (Figure 2A). Since IgA-mediated lysis is most effective with medium to high levels of antigen expression, >200,000 molecules per cell, we determined antigen density of the PDAC cell lines (Figure 2C) [47,48]. EpCAM expression exceeded 200,000 molecules per cell on all cell lines except for BxPC-3, while for TROP2 antigen expression was only above 200,000 for BxPC-3. EGFR had even lower antigen expression (<150,000 molecules) on all cell lines, suggesting that EpCAM might be the most suitable target for IgA immunotherapy of PDAC (Figure 2C). Screening for mucins showed that MUC1, but not MUC4 and MUC16, is expressed on PDAC cell lines (Figure 2B and Appendix A). MUC1 expression was relatively lower on AsPC-1 and BxPC-3 when compared to the other cell lines (Figure 2B). Expression of the tumor-associated (TA) MUC1 TN/STn carbohydrate epitopes was tested, but only low expression was found on Capan-2 and CFPAC-1 (Appendix A). Since these MUC antibodies have multiple binding sites on the same molecule, it is not possible to determine absolute antigen expression levels.

Since EpCAM and TROP2 showed the highest antigen density, IgA mAbs against these targets (heING1 for EpCAM and sacituzumab for TROP2) were evaluated against their IgG counterparts in WL ADCC against the PDAC cell lines Panc 10.05 and Capan-2 (Figure 2D,E and Appendix A). The highest concentration of antibody without effector cells (Ab only) showed no lysis (Figure 2D–H), demonstrating that IgA binding to a TAA does not affect viability in this short-term assay. In addition, no killing was observed with effector cells in absence of antibody (0) (Figure 2D–H). The same controls were used for cells treated with CD47 blocking antibody, showing that CD47 blockade alone does not induce lysis or reduce viability (Figure 2D–H). Lysis of Panc 10.05 was low for both the IgA and IgG mAbs (Figure 2D). CD47 blockade improved lysis by eightfold for EpCAM-IgA and by fivefold for EpCAM-IgG and overall lysis was significantly higher with IgA than IgG mAbs when used in combination therapy (Figure 2D). Similar results were observed with TROP2 mAbs (Figure 2E). IgG was somewhat more effective than IgA in ADCC against Capan-2 cells both with and without addition of CD47 blockade, although differences in lysis were small and a maximal lysis of only 7.6% was achieved (Appendix A).

Following these results, we tested IgA mAbs against EpCAM, EGFR (cetuximab) and TROP2 on PDAC cell lines in ADCC with PMNs (mainly comprising of neutrophils). IgA antibodies showed no efficacy against AsPC-1 and BxPC-3 cell lines, even after disruption of the CD47/SIRPα-axis (Appendix A). The EpCAM-IgA mAb was able to induce lysis of the Panc 10.05 cell line while EGFR-IgA and TROP2-IgA were not effective (Figure 2F). Addition of CD47 blockade enhanced lysis of the highest concentration EpCAM-IgA by fourfold and TROP2-IgA by 18 fold, while EGFR-IgA remained ineffective (Figure 2F). Antibody efficacy showed a similar pattern for Capan-2 and CFPAC-1, although lysis was slightly lower (Figure 2G,H). For Capan-2, CD47 blockade enhanced lysis by threefold for EpCAM-IgA and sixfold for TROP2-IgA (Figure 2G). Lysis of CFPAC-1 increased with CD47 blockade by threefold for EpCAM-IgA and by ninefold for TROP2-IgA (Figure 2H). EGFR-IgA killing remained ineffective with CD47 blockade (Figure 2G,H).

### 3.2. Patient Tumors Express Similar TAAs as PDAC Cell Lines

Tumor material was obtained from 25 PDAC patients after surgical resection and dissociated for analysis (Figure 1). The size of the extracted tumor material differed per patient and ranged from roughly 0.5–2.5 cm^3^. Fourteen out of the 25 tumor samples were included in the study since dissociation of tumor tissue or antibody staining was not always successful due to high fibrosis, which can result in cell death due to the aggressive dissociation method required, or lack of isolated tumor cells. Additionally, three samples resected for suspicion of PDAC were diagnosed as other (pre)malignancies after pathological assessment (two times cholangiocarcinoma and one intraductal papillary mucinous neoplasm) and were therefore excluded. The cells extracted from these 14 tumor samples ranged between 300,000 and 15 million (Table 1). TAA staining was performed on all samples, while ADCC assays and immunophenotyping were done when sufficient cells could be extracted. Leukocyte composition was evaluated on four samples and ADCC was done with tumor material from two patients (Table 1). When assessing the TAA expression on the tumor material, it was not possible to exclude all fibroblasts using flow cytometry analysis due to the lack of a general fibroblast marker (Appendix A). Therefore, the CD45 negative population contains both tumor cells as well as fibroblast, and will probably cause an underestimation of the number of TAA positive cells (Appendix A). To be able to assess TAA expression on tumor cells specifically, we assumed all tumor cells expressed the epithelial marker EpCAM and determined the percentage of TAA expressing cells relative to the EpCAM positive population (Figure 3A). In line with TAA expression on PDAC cell lines, EpCAM, TROP2 and MUC1 were the most abundant expressed TAA on patient tumor cells (Figure 3A,B and Appendix A). EpCAM was expressed on 12/14, EGFR on 6/12, TROP2 on 10/12 and MUC1 on 13/14 of the tumors (Figure 3C). EGFR was expressed in 50% of the patients, although expression was low in 5/6 samples (Figure 3). TROP2 and CD47 were later inclusions in the panel and not analyzed in the first two patient samples (Figure 3).

### 3.3. Tumor Cells Isolated from Patient PDAC Can Be Killed by Neutrophils with IgA mAbs

Since EpCAM was the most abundant and consistent expressed TAA on PDAC patient tumor cells, the EpCAM-IgA antibody was tested in PMN ADCC assays with tumor cells and compared to PDAC cell lines. In two patient tumor samples, sufficient cells were isolated to perform ADCC assay. The cell isolated from patient sample #10 showed 80% EpCAM positivity after tumor cell purification using negative MACS selection (Figure 4A). Since not all fibroblasts could be excluded, we did not expect 100% positivity and we corrected lysis for EpCAM positivity. The highest IgA mAb concentration induced 6.2% lysis of the patient tumor cells (by PMN) and 3.6% of the Panc 10.05 cell line. Addition of CD47 blockade enhanced lysis of the highest mAb concentration by 2.3 fold for the tumor cells and sixfold for Panc 10.05 (Figure 4B). The EpCAM positivity of patient sample #14 was 70% after tumor cell purification (Figure 4C). Patient tumor cells were only killed with addition of CD47 blockade in this sample (Figure 4D). Upon CD47 block, killing of the tumor cells was 11.4% and 11.6% for the Capan-2 cell line (Figure 4D). Four percent lysis was achieved in a third tumor sample, which could be enhanced to 14% by addition of CD47 blockade. However, this tumor was later diagnosed as cholangiocarcinoma (Appendix A). These data together show that IgA-EpCAM antibodies can kill PDAC patient tumor cells ex vivo, emphasizing the rationale to continue testing IgA therapy, in combination with CD47 blockade, for treatment of PDAC.

### 3.4. Cell Types and Immune Cells in Patient Tumor Samples

To gain a better insight into the cellular composition of the PDAC patient tumors, immune cells were characterized in four patients. Patient tumor samples consisted of RBCs, leukocytes, epithelial cells (tumor cells) and an unidentified fraction of other cells (Figure 5A and Appendix A). The infiltrations of immune cell subsets differed between tumor samples (Figure 5B and Appendix A). PMNs were found in all tumor samples (Figure 5B). Almost all PMNs showed expression of the Fc alpha receptor (FcαR), which is required for IgA immunotherapy (Figure 5C). Furthermore, the FcαR is also expressed on the majority of monocytes, indicating that these cells could also be activated by IgA mAbs (Appendix A). LOX-1 expression on PMN implies that an immunosuppressive subset of neutrophils was present in the TME (Figure 5D). Further analysis of T cells subsets showed that both CD4+ and CD8+ T cells were present in the tumor samples (Appendix A). These results demonstrate that the tumors contain all immune cell populations which can be activated by IgA mAbs. In combination with the abundance of EpCAM on PDAC patient material and the observation that tumor cells can be killed ex vivo by neutrophils upon IgA-EpCAM treatment in combination with CD47 blockade, we provide a basis for further exploration of IgA-EpCAM antibodies for the treatment of this therapy resistant disease.

## 4. Discussion

In this study, we identified EpCAM, TROP2, and MUC1 as potential targets for (IgA) mAb therapy in PDAC. The TAAs EpCAM, TROP2, and MUC1 showed the highest and most consistent expression across five PDAC cell lines and 14 primary tumor samples. For EpCAM and TROP2, we were able to induce ADCC with IgA mAbs in combination with CD47 blockade in three out of five cell lines. In addition, we observed a similar pattern of lysis in two primary (patient) tumor cell samples with EpCAM IgA mAb (with CD47 blockade). We identified MUC1 as a third potential target based on expression levels in PDAC, but we were not successful in generating an effective MUC1 IgA mAb. We have shown that IgA mAbs could be an effective alternative to IgG mAbs, especially since PDAC tumors are characterized by predominant infiltration of neutrophils and macrophages [17,18,19]. Furthermore, it should be mentioned that NK cells, which are important effector cells for IgG, are often suppressed in the PDAC TME [19,25].

EpCAM was one of the first tumor-associated antigens to be described [49]. Although it is overexpressed in many epithelial cancers, it is also expressed on a wide range of healthy epithelia [49,50]. Phase I and II clinical trials have shown that EpCAM IgG mAbs can be safely administered for the treatment of EpCAM-overexpressing adenocarcinomas [51,52,53]. However, acute pancreatitis may be a concern with regard to high-affinity EpCAM mAbs, such as heING1 [52,53]. IgA mAbs seem to require a certain threshold of target expression for efficient ADCC induction, which could be an advantage when compared with IgG mAbs [47,48]. This threshold may spare low-expressing normal epithelial cells, but also tumor cells that lose antigen expression. According to data from the Human Protein Atlas, EpCAM mRNA expression is higher in PDAC than of normal pancreas [54,55]. Moreover, EpCAM expression in/of normal pancreas and PDAC has been compared via immunohistochemistry, showing an overexpression of EpCAM in 56–63% of PDAC tumors and higher expression relative to normal pancreas [56,57]. The EpCAM expression of normal pancreas epithelium should be quantified and compared to tumor cells in ADCC to determine the viability of EpCAM as a mAb target. In our experimental setup, lysis differed per cell line and not all cell lines could be killed by EpCAM-IgA in combination with CD47 blockade (Figure 2 and Appendix A). These differences in efficacy between cell lines may only be partly explained by antigen density. Other mechanisms of resistance, such as immune checkpoints other than CD47, might play a role and should be further studied [58,59]. The clinical efficacy of EpCAM IgG mAbs in breast and colon carcinoma is limited [49]. Since IgG mAbs predominantly activate different immune cells, such as NK cells, it should be further evaluated whether IgA EpCAM mAbs can pose an alternative strategy against PDAC and cancer in general.

TROP2 is frequently overexpressed in epithelial cancers and is expressed in various healthy tissues [46]. In line with our findings, TROP2 has been described in primary and metastatic PDAC [46]. In recent years TROP2 has become an established therapeutic target with the development of an antibody-drug conjugate sacituzumab govitecan for the treatment of triple-negative breast cancers [60,61]. In our study, we used the IgG and IgA formats of sacituzumab and showed ADCC with addition of CD47 blockade, despite relatively low antigen density at 90,000 per cell for Panc 10.05. This efficacy at lower antigen densities cannot be explained by higher antibody affinity, since the sacituzumab antibody has a slightly higher K_D_ value compared with the other antibodies (heING 0.16 nM, cetuximab 0.20 nM and sacitizumab 0.26 nM) [53,62,63]. Some targets may be more suitable for mAb targeting due to a better binding site, differences in size (closer binding to the membrane to exert trogocytosis), or because they are not internalized after binding, which improves ADCC [64,65]. It is important to note that a TROP2 antibody of the NY18 clone did not bind to Capan-2, implying that some of the tumor samples could have been false negatives. As such, we switched to the MR54 TROP2 antibody for antigen density determination. TROP2 can express TA epitopes in cancer, which can explain differences in antibody binding [66]. Antibodies against these TA epitopes have been developed and their expression should be tested on PDAC [66]. If expressed, targeting these epitopes with IgA mAbs could offer a more selective therapeutic approach, although antigen density might be a limitation.

IgA mAb against EGFR were ineffective against PDAC cell lines, suggesting that the antigen density on PDAC cell lines (<150,000) was too low to induce ADCC with neutrophils, especially considering that EGFR IgA mAbs are effective against other cancer cell lines [39]. In line with this, we found low expression of EGFR on patient tumor cells, which could explain the inefficacy of IgG mAbs in clinical trials for PDAC [15,16]. In addition to Fc-mediated immune cell activation, EGFR mAbs are also applied to block the EGFR signaling pathway in cancer. Downstream activation of the EGFR signaling pathway, such as KRAS mutations present in 90% of PDACs, can impair efficacy and could further explain the inefficacy of EGFR mAbs observed in clinical trials for PDAC [67,68].

Mucins are large, heavily glycosylated proteins expressed on the membrane of epithelial cells. Mucins, specifically MUC1, are overexpressed in cancer, and MUC1 expresses aberrantly glycosylated TA epitopes in 80% of PDAC patients [69,70,71]. High MUC1 expression in PDAC is associated with poorer overall survival [71,72]. What makes MUC1 a particularly interesting target for IgA mAbs is its tumor-specific expression of the sialyated core 1 glycan (MUC1-ST), which interacts with Siglec-9 on myeloid cells [73]. The sialoglycan/Siglec-9 axis can function as a myeloid immune checkpoint and disruption of this axis can improve PMN ADCC [58,74]. In addition, MUC1 can interact with ErbB receptors, including EGFR, and drive growth factor signaling [75]. These findings demonstrate that therapies targeting MUC1 could also inhibit the EGFR signaling pathway. In line with the literature, we found MUC1 expression on PDAC. Expression of the TA Tn/STn carbohydrate epitopes seemed to be very low in our samples, and targeting these epitopes with IgA did not result in ADCC by neutrophils (Appendix A). MUC1 epitopes may differ between patient tumors and analysis of different epitopes is necessary. MUC1 mAbs have been ineffective in clinical trials and it is not clear whether MUC1 is a suitable target for targeted mAb therapy [76]. Challenges concerning MUC1 targeting might involve its relatively large size, glycosylation, and the associated barrier properties. These characteristics could interfere with IgA mAb efficacy, since neutrophils activated by IgA need to be able to latch onto the membrane to perform trogocytosis and induce tumor cell killing. In addition, MUC1 mAb targeting of glycoepitopes could result in only the removal of these carbohydrates, similar to the removal of antigens via shaving [77,78]. MUC1 mAbs against TA epitopes pose an interesting option to selectively target the cancer cells in PDAC; whether IgA mAbs can effectively target MUC1 in cancer should be further evaluated.

We showed that blockade of the CD47/SIRPα myeloid immune checkpoint can enhance PMN ADCC and was necessary to induce efficient lysis. However, CD47 is also expressed by healthy tissue and is highly expressed on red blood cells to protect them from phagocytosis [79,80]. Although CD47 is overexpressed on PDAC tumor cells compared to adjacent normal epithelium [81], blockade of CD47 could pose a risk for auto-immunity such as hemolysis [80,82,83]. Blockade of SIRPα on neutrophils/myeloid cells could be an alternative strategy but can still result in auto-immunity [80]. Another solution could be the generation of bispecific antibodies with one antigen binding site targeting a TAA and the other binding to CD47 with lower affinity, preventing effective CD47 blockade in absence of the TAA [80,82]. A different strategy could be an antibody with pH-dependent antigen binding in which TAA or CD47 binding is specific to the acidic TME of PDAC [84]. However, this strategy does impair the ability to target circulating tumor cells, which poses a major issue for the treatment of PDAC since mortality can mainly be attributed to metastatic disease. As previously mentioned, the role of other immune checkpoints such as Siglecs should be considered. Siglec-7 and Siglec-9 on myeloid cells can interact with sialoglycans on tumor cells as well as cancer-associated fibroblasts in PDAC, and blocking Siglec-7 and Siglec-9 can enhance PMN ADCC [58,85,86,87].

We used 51Cr release assays to measure ADCC. These assays require target cells to take up 51Cr for labeling, which can vary per cell line and can also be hampered by lower viability [88]. This might influence the levels of lysis that can be measured, especially in patient tumor cells, where viability quickly decreases in vitro after tumor dissociation, reflected by lower CPM in maximal lysis. Since 51Cr is toxic for cells, the viability of vulnerable cells can be further reduced [88]. It remains unclear what levels of lysis translate to therapy efficacy in vivo. Initiation of some lysis by neutrophils will attract more neutrophils and could initiate a larger adaptive immune response against the tumor cells [89]. We used negative selection to isolate the tumor cells for ADCC; however, this method can be improved by the addition of more fibroblast markers to the depletion panel, such as fibroblast activator protein and alpha smooth muscle actin, if these are not expressed on tumor cells [17]. In this study, we used PMN from healthy donors; previous results showed that PMN isolated from peripheral blood of PDAC patients can induce similar levels of ADCC with IgA mAb against PDAC cell lines [58].

Another limitation of in vitro ADCC assay is that it does not take the TME into account. As mentioned before, cancer-associated fibroblasts can inhibit anti-tumor responses [58,87]. Furthermore, the TME of PDAC is characterized by a dense desmoplastic stroma which can make up 80–90% of the tumor mass [3,21]. This stroma often prevents interaction between immune cells and tumor cells [90]. It is also responsible for poor perfusions of PDAC tumors, causing a hypoxic, suppressive TME that also makes drug delivery via systemic therapy challenging [91]. It is sometimes reported that certain immune cells are absent from PDAC tumors altogether [92]. The leukocyte/RBC ratio in blood is 1:600, while in our samples, the average ratio was 1:0.6. This suggests that most leukocytes measured were present in the tumor and not due to spillage from the vasculature. More relevant for IgA mAb therapy is that FcαR-expressing neutrophils and monocytes were present. The tumor digestion process can decrease certain immune cell fractions, specifically cells such as neutrophils, which are sensitive and quickly go into apoptosis [93]. For proper assessment, the TME of PDAC should be further evaluated via different methods. Techniques such as imaging mass cytometry in combination with cytometry by time of flight can help to better map the immune micro-environment of PDAC by identifying different immune cell subsets, their phenotypes and functions, and their localization in relation to tumor cells [17,94,95].

## 5. Conclusions

The development of new treatments for PDAC is challenging. Immunotherapies are mostly ineffective due to the highly immunosuppressive TME, which includes immunosuppressive neutrophils. Research efforts are increasingly focusing on specific targeting to deplete or reduce neutrophils, with the aim to improve the efficacy of anti-cancer therapies. However, neutrophils can also be manipulated to exhibit anti-tumor effects using IgA mAbs. Here, we show that IgA antibodies targeting EpCAM and TROP2 in combination with CD47 blockade may represent a promising therapeutic strategy for PDAC in the preclinical setting, which merits further investigation.

## Figures and Tables

**Figure 1 cells-14-00632-f001:**
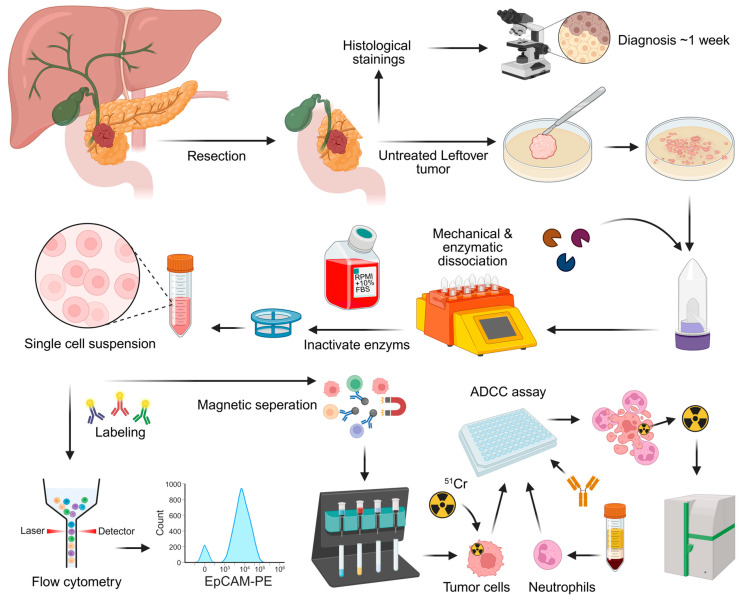
Processing of patient tumors.

**Figure 2 cells-14-00632-f002:**
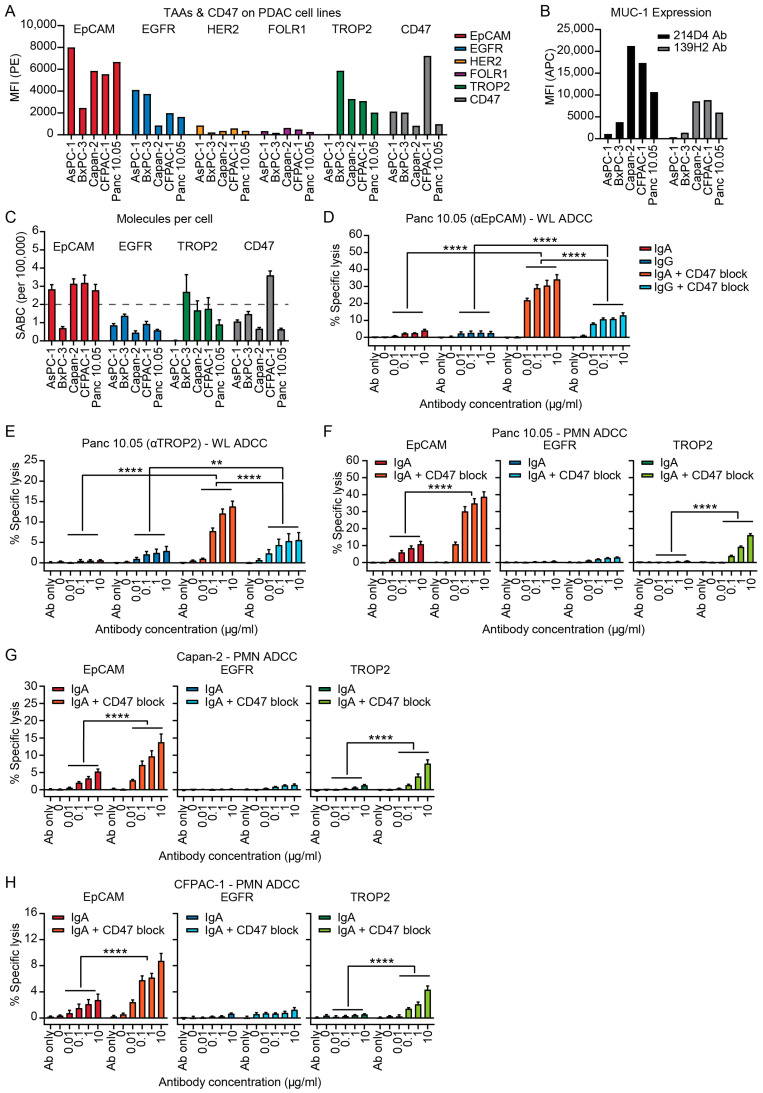
Tumor-associated antigens (TAAs) expressed on pancreatic ductal adenocarcinoma (PDAC) cell lines and efficacy of IgA monoclonal antibodies (mAb) targeting these TAAs in antibody-dependent cellular cytotoxicity (ADCC) assay. (**A**) Mean fluorescent intensity (MFI) of TAA and CD47 expression as measured with flow cytometry. Indirect staining was performed with 10 µg/mL target IgA antibody and for CD47 a PE-conjugated antibody was used. 3 independent experiments are shown in 1 representative graph. (**B**) MFI of mucin-1 (MUC1) expression as measured with flow cytometry. Indirect antibody staining was performed with 10 µg/mL 214D4 antibody (Ab) or 10 µg/mL 139H2 Ab. 3 independent experiments are shown in 1 representative graph. (**C**) Antigen density of EGFR, EpCAM, TROP2 and CD47 was determined with QIFIKIT analysis using 10 µg/mL target antibody. Mean ± standard error of the mean (SEM) of specific antibody binding capacity (SABC) is shown of a minimum of n = 3 independent experiments. (**D**–**H**) Tumor cells were labeled with chromium-51 (51Cr) for chromium release assays and subsequent chromium release was measured after 4 h of incubation with antibodies and whole leukocytes (WL) or polymorphonuclear leukocytes (PMN). For CD47 block, target cells were incubated with IgG1 PGLALA SIRPα fusion protein prior to 51Cr release ADCC assay. (**D**) Panc 10.05 ADCC with WL comparing IgA with IgG mAb against EpCAM and (**E**) against TROP2. (**F**) PMN ADCC of Panc 10.05, (**G**) Capan-2 and (**H**) CFPAC-1 cell lines with IgA mAb targeting EpCAM (heING1), EGFR (cetuximab) and TROP2 (sacituzumab). The Mean ± SEM of specific lysis is shown of at least n = 3 independent experiments. Independent experiments are performed in technical triplicate. Two-way ANOVA followed by Tukey’s post-hoc test was performed. ** *p* < 0.01, **** *p* < 0.0001.

**Figure 3 cells-14-00632-f003:**
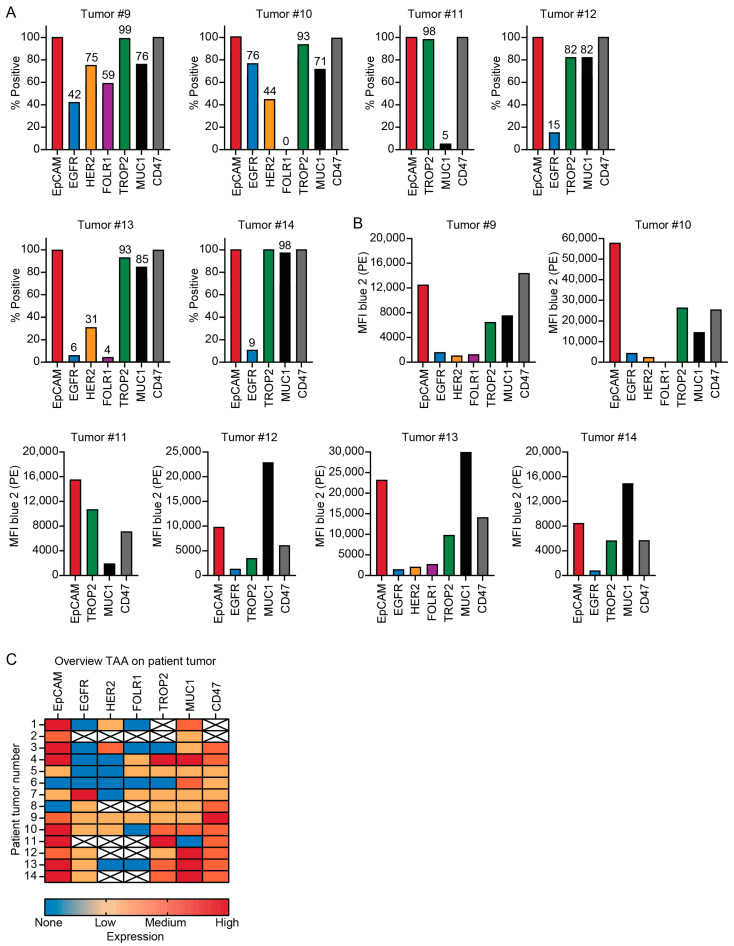
Tumor-associated antigens (TAAs) and CD47 expression on tumors from patients with pancreatic ductal adenocarcinoma (PDAC). Digested tumor samples were measured with flow cytometry after RBC lysis. Dead cells (To-Pro-3) and leukocytes (CD45+) were excluded. Exclusion of all fibroblasts was not possible and we assumed all tumor cells expressed the epithelial marker EpCAM. (**A**) Bar graphs showing the percentage of TAA positive cells relative to EpCAM. (**B**) Bar graphs depicting the mean fluorescent intensity (MFI) of the TAA positive cell populations. (**C**) Heatmap depicting the relative expression (based on MFI) of different TAAs and CD47 on PDAC patient tumor cells, red is high expression, orange is low expression and blue is no expression. The TAAs depicted in white were not determined. The targets are represented as measured, however low viability and cell numbers can influence the results and cause false negatives.

**Figure 4 cells-14-00632-f004:**
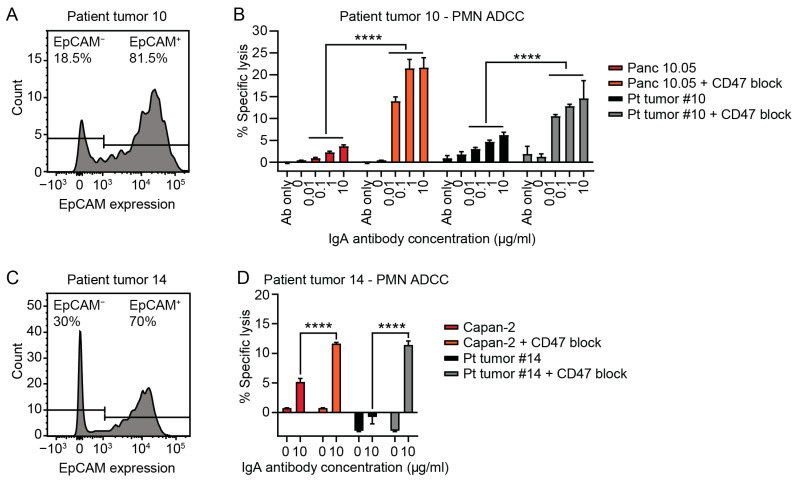
Efficacy of EpCAM-IgA against patient tumor cells. Polymorhonuclear leukocyte (PMN) antibody-dependent cellular cytotoxicity (ADCC) assay with IgA against EpCAM and pancreatic ductal adenocarcinoma (PDAC) tumor cells isolated via negative MACS selection (CD45, CD31, PDGFRα- and PDGFRβ negative) after red blood cell lysis. Lysis of patient tumor cells was corrected to percentage of isolated tumor cells since not all fibroblast could be excluded from the sample. EpCAM expression in (**A**) tumor sample #10 and (**C**) #14 after tumor cell isolation. ADCC of Panc 10.05 cell line and tumor cells from (**B**) patient sample #10 and (**D**) #14. The mean ± SEM of specific lysis of a technical triplicate is shown. Two-way ANOVA followed by Tukey’s post-hoc test was performed. **** *p* < 0.0001.

**Figure 5 cells-14-00632-f005:**
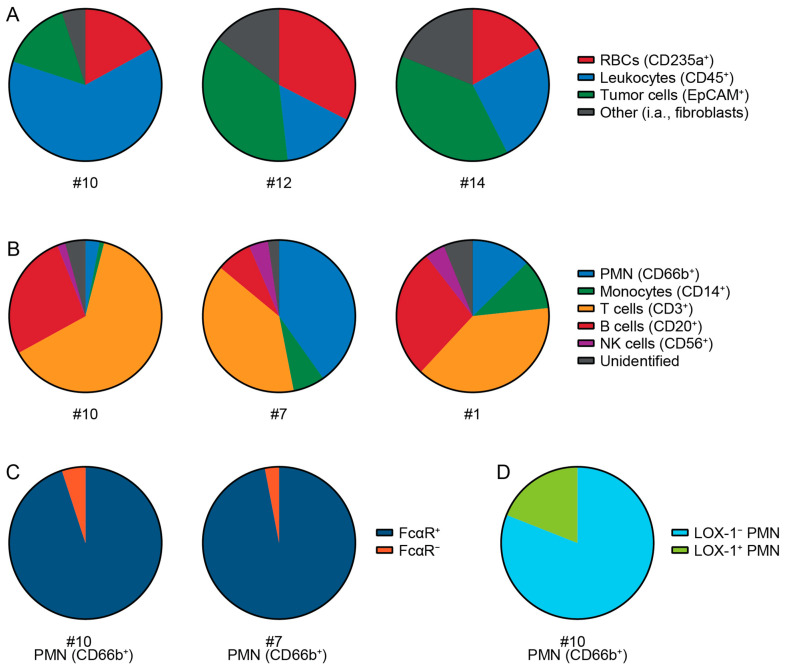
Characterization of cell types in patient tumor samples Patient tumors were digested to a single cell suspension and further analyzed with flow cytometry. (**A**) Patient tumor samples contain red blood cells (RBCs, CD235a), leukocytes (CD45) and tumor or epithelial cells (EpCAM). For further analysis red blood cell lysis was performed. Dead cells were excluded with 7-AAD as live-death marker. (**B**) Leukocyte composition evaluation in tumor sample shows the presence of granulocytes (PMN), monocytes, T, B, and NK cells. (**C**) Staining for the Fc alpha receptor (FcαR) and (**D**) LOX-1 was performed on PMN.

**Table 1 cells-14-00632-t001:** Included patient tumor samples.

Patient Tumor	Cells Extracted (×10^6^)	TAA Staining	Leukocyte Composition Evaluation	ADCC Assay
1	2.5	+	+	
2	0.75	+		
3	2.2	+		
4	15	+		
5	3.0	+	+	
6	1.9	+		
7	3.8	+	+	
8	0.80	+		
9	0.30	+		
10	4.9	+	+	+
11	0.80	+		
12	0.50	+		
13	0.42	+		
14	1.2	+		+

TAA = tumor-associated antigen, ADCC = antibody-dependent cellular cytotoxicity.

## Data Availability

The research data are stored in an institutional repository and will be made available by the authors on request. Further information and requests for resources and reagents should be directed to and will be fulfilled by the lead contact, J.H.W.L.

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
