# Peer review of "The Efficacy of Targeted Monoclonal IgA Antibodies Against Pancreatic Ductal Adenocarcinoma"

_cells, 2025, doi:10.3390/cells14090632_

Round 1
Reviewer 1 Report
Comments and Suggestions for Authors
In this study, the authors clearly presented their work on detecting the efficacy of IgA antibodies against PDAC. The study design is reasonable, and the results support the conclusions. A few questions remain to be answered:
- In section 3.2, please explain the reason of comparing percentage of TAA relative to EpCAM (Figure 3A). It is not likely that EpCAM is expressed consistently in each patient sample based on the results of Figure 3B.
- In section 3.3, It is necessary to explain to the reason of selecting patient samples 10 and 14 in ADCC assay.
- In section 3.3, if other cell lines are tested for CD47 blocking in comparison with patient tumor 14, what do the results look like?
- Blocking of CD47 seems to be more effective than IgA antibodies from the results of ADCC, are the comparisons between IgA vs. CD47 blocking alone performed?
Reviewer 2 Report
Comments and Suggestions for Authors
The study is scientifically rigorous and presents novel therapeutic insights. With improved clarity and expanded discussion on translational challenges, it will make a strong contribution to immunotherapy research in PDAC.
- Add quantification of EpCAM/CD47 expression in normal pancreatic tissues to evaluate therapeutic windows.
- Include more details on antibody affinity (e.g., KD values) if available, to support differences in efficacy.
- Explore potential resistance mechanisms (e.g., stromal barriers, alternative immune checkpoints).
- Consider testing bispecific or pH-responsive antibodies as discussed, in future directions or as speculative outlook.
- Improve figure readability, possibly by separating dense panels or highlighting key comparisons.
- Possible toxicity of CD47 blockade.
- Figures are informative but could benefit from clearer legends (some are overly dense).
- The authors acknowledge fibroblast contamination, but additional quantification of tumor cell purity would be helpful.
- Consider adding a brief rationale for choosing EpCAM, TROP2, and MUC1 from the initial antigen panel.
Round 2
Reviewer 2 Report
Comments and Suggestions for Authors
we satisfied with the revised manuscript.